# Pleiotropic Roles of the Orthologue of the *Drosophila melanogaster Intersex* Gene in the Brown Planthopper

**DOI:** 10.3390/genes12030379

**Published:** 2021-03-07

**Authors:** Hou-Hong Zhang, Yu-Cheng Xie, Han-Jing Li, Ji-Chong Zhuo, Chuan-Xi Zhang

**Affiliations:** 1Institute of Insect Science, Zhejiang University, Hangzhou 310058, China; zhanghouhong@zju.edu.cn (H.-H.Z.); 21816103@zju.edu.cn (Y.-C.X.); 21916101@zju.edu.cn (H.-J.L.); 2State Key Laboratory for Managing Biotic and Chemical Threats to the Quality and Safety of Agro-Products, Key Laboratory of Biotechnology in Plant Protection of MOA of China and Zhejiang Province, Institute of Plant Virology, Ningbo University, Ningbo 315211, China; zhuojichong@nbu.edu.cn

**Keywords:** *intersex*, RNA interference, external genital, copulatory bursas, reproductive system, *Nilaparvata lugens*

## Abstract

*Intersex*(*ix*), a gene involved in the sex-determining cascade of *Drosophila melanogaster*, works in concert with the female-specific product of *doublesex* (*dsx*) at the end of the hierarchy to implement the sex-specific differentiation of sexually dimorphic characters in female individuals. In this study, the *ix* homolog was identified in the brown planthopper (BPH), *Nilaparvata lugens*, which contained two splice variants expressed in both female and male insects. We found that *Nlix* played a vital role in the early nymphal development of BPH, showing an accumulated effect. RNAi-mediated knockdown of *Nlix* at 4th instar led to the external genital defects in both sexes, consequently resulting in the loss of reproductive ability in female and male individuals. After dsRNA injection, the males were normal on testes, while the females had defective ovarian development. *Nlix* was also required for early embryogenesis. Notably, when the ds*Nlix* microinjection was performed in newly emerged females, the copulatory bursas were abnormally enlarged while the other tissues of the reproductive system developed normally. Our results demonstrated the pleiotropic roles of *Nlix* in embryogenesis and development of the reproductive system in a hemimetabolous insect species.

## 1. Introduction

With the evolution of organisms, the sex-determining mechanisms presented an astounding diversity in insects, which have affected the sexually dimorphic development at the morphological, behavioral and physiological levels. *Intersex* (*ix*) is a gene involved in the sex-determining cascade of insects. It was first well-characterized in *Drosophila melanogaster* [1] and showed apparently deep conservation in insect species, as *Bombyx mori* [2], *Maruca vitrata* [3], *Cyclommatus metallifer* [4], *Bemisia tabaci* [5] and *Oncopeltus fasciatus* [6]. In *D. melanogaster*, the DmIX protein forms a complex with the female-specific product of *Dmdsx* (DmDSX^F^) at the bottom of the sex-determining cascade, thereby implementing the sex-specific differentiation of sexually dimorphic characters among female individuals [1,7,8]. In contrast, the *ix* functioned independently of *dsx* in *B. mori*, and the phenotype of *Bmix* mutants led to irregular development of external genitalia in female insects while leaving the external genitalia of male individuals unaffected. Apart from sexual features, developmental abnormalities were also found in the imaginal disc, including leg, wing and antenna, of both female and male silkworm mutants [2]. Furthermore, the intersexual characteristics of the sexual somatic structures in *D. melanogaster* females resulting from the absence of *Dmix* could be partially rescued with the expression of *ix* homologs from *B. mori* and *M. vitrata* [3,7,9]. In *O. fasciatus* and *C. metallifer*, knockdown of *ix* gene in females resulted in masculinizing phenotypes as reported in *D. melanogaster*. RNAi knockdown of *ix* led to reduced genitalia size in *O. fasciatus* males, while exerting indistinct effects in *C. metallifer* males [4,6]. Besides, the *ix* homolog of *B. tabaci* has been confirmed to produce four isoforms by alternative splicing, including one female splice variant, one male splice variant and two non-sex-specific unspliced variants in contrast to *ix* of *D. melanogaster* [5]. In the RNAi experiments of whitefly adults, no distinction in the morphological structure of genitalia was detected between wildtype and RNAi-treated groups, despite somewhat reduced fecundity and eclosion rate of progeny in females.

The brown planthopper (BPH), *Nilaparvata lugens* Stål (Hemiptera: Delphacidae), a representative hemimetabolous insect, is one of the most devastating rice pests in many Asian countries [10]. To date, scarce details of molecular mechanism for sex determination in BPH have been understood, with the only described genes of the sex-determining pathway being *transformer 2* (*Nltra2*) [11] and *doublesex* (*Nldsx*) [12]. 

It has been indicated that the phenotypes of *ix*-depleted gene corresponded to the phenotypes associated with *dsx*-depleted gene in *D. melanogaster* females [1,8]. Loss of *dsx* function would transform females and males into intersexual phenotype in *D. melanogaster* [12], while among BPHs, the loss of *dsx* function only resulted in males to feminization [13]. Considering the phenotypic divergence between the above two species induced by *dsx* depletion, it is necessary to further explore the function and characteristics of *ix* in BPH. Our results revealed that the *ix* gene was essential for the development, reproduction and embryogenesis of BPH. Unlike *D. melanogaster*, defective phenotypes of *ix* silencing were found in both BPH sexes. 

## 2. Materials and Methods

### 2.1. Insect Rearing

The BPH population used in this study was originally collected from local rice fields in Hangzhou, China, in 2008. The insects were reared on rice seedlings (variety: Xiushui 134) in a greenhouse set at 26 ± 1 °C with 60–70% relative humidity under a 16 h light: 8 h dark photoperiod.

### 2.2. Searching for Ix Homologs in BPHs and Sequence Analysis

DmIX protein sequence obtained from the NCBI database was used as the tblastn query sequence for searching IX homologs in local BPH genomic and transcriptomic databases with a cut-off e-value <10^−5^. The transcriptomic homolog sequences of *Nlix* were aligned with the corresponding genomic sequences by using the Splign tool on the NCBI website (www.ncbi.nlm.nih.gov/sutils/splign/splign.cgi, accessed on 5 December 2020), in order to identify the introns, exons and intron-exon boundaries. These splicing structures and predicted open reading frames (ORFs) were confirmed by PCR. Appendix A lists the primers used in PCR.

### 2.3. Sequence Comparison and Phylogenetic Relationships

The ORF and amino acid sequence of *Nlix* gene were predicted using the Translation tool on the ExPASy Portal (http://web.expasy.org/translate/, accessed on 5 December 2020). The putative protein domains were deduced utilizing the NCBI conserved domain search program (https://www.ncbi.nlm.nih.gov/Structure/cdd/wrpsb.cgi, accessed on 5 December 2020). ClustalW program was used to make sequence comparisons. A phylogenetic tree among the IX proteins was computed by a neighbor-joining algorithm using the MEGA X software. Phylogenetic relationships were determined using 1000 bootstrap replicates and the branches with a value less than 50% were collapsed. In this study, the sequences that have been identified from *Coptotermes formosanus, Zootermopsis nevadensis, Cryptotermes secundus, Oncopeltus fasciatus, Bemisia tabaci, Halyomorpha halys, Cimex lectularius, Melanaphis sacchari, Aphis gossypii, Sipha flava, Thrips palmi, Frankliniella occidentalis, Trypoxylus dichotomus, Tribolium castaneum, Photinus pyralis, Sitophilus oryzae, B. mori, B. mandarina, Papilio polytes, Maruca vitrata, Helicoverpa armigera, D. melanogaster, D. virilis, Culex quinquefasciatus, Megaselia scalaris, Anopheles darlingi, Apis cerana, Dufourea novaeangliae, Bombus impatiens, Megalopta genalis, and Osmia lignaria* were analyzed. Appendix A provides the GenBank accession numbers of the sequences used in this analysis.

### 2.4. Expression Pattern Analysis of Nlix

Developmental stage samples were obtained from BPHs in different stages, which included 12 egg samples, 5 nymph samples and 2 adult samples collected every 24 h following molting/emergence. Different tissue samples, including fat body, integument, gut, ovary and testis were dissected from adult females and males 72 h after emergence. Total RNAs were extracted using RNAiso Plus (Takara, Kyoto, Japan). Subsequently, 1 μg of total RNA treated with DNase I was used to perform reverse transcription and first-strand cDNA synthesis in 10 μL of reactions, which was accomplished using the HiScript® II QRT SuperMix (Vazyme, Nanjing, China) according to the manufacturer’s instructions. Real-time qPCR (RT-qPCR) was conducted to quantify the target gene using a Bio-Rad Real-time PCR system (BioRad, Hercules, CA, USA). Gene-specific primers were designed using the Primer Premier 6 program (Appendix A). The 18S ribosomal RNA (*Nl18S*) (GenBank accession number JN662398.1) was used as an internal control. The quantitative variations were evaluated by 2^ΔΔCt^ relative quantification method(2^ΔΔCt^) [14]. Three biological replications were conducted per sample.

### 2.5. RNAi Interference

The double-stranded RNAs of the target gene were synthesized by in vitro transcription with the purified DNA templates using a T7 High Yield RNA Transcription Kit (Vazyme, Nanjing, China), according to the manufacturer’s instructions. Two unique regions of the *Nlix* gene were used to synthesize the dsRNAs, respectively. The green fluorescent protein of *Aequorea victoria* was used as a negative control. Appendix A lists the primers used for dsRNAs synthesis. Microinjection of BPH with dsRNA was performed as reported previously [15,16]. In all experiments, 150 nymphs 3rd (day 1) and 4th (day 1) instars were used for dsRNA treatment, and each treatment was performed in three independent biological replications. Each instar nymph was injected with 10 nL of dsRNA (5 ng/μL) into the mesothorax using the FemtoJet Microinjection System (Eppendorf-Netheler-Hinz, Hamburg, Germany). Three days after injection by RT-qPCR, 6 to 10 insects were collected as one independent sample to evaluate the RNAi efficiency of *Nlix*. The other insects were used for developmental observation and survival quantification.

### 2.6. Fertility Analysis

To determine the silencing effect of ds*Nlix* on BPH fecundity and hatchability, we performed the microinjection on late 5th instar males and newly emerged females. Three days after emergence, the BPH pairs, either a healthy male & a treated female or a treated male and a healthy female, were transferred into glass tubes containing three fresh rice seedlings to oviposit for five days. Subsequently, the insects were removed, and the rice seedlings were maintained for another ten days for counting the number of hatched offspring. Afterward, the leaf sheaths of the rice seedling were dissected under the microscope to count the number of eggs that failed to hatch. Each treatment was carried out in fifteen biological replications.

### 2.7. Statistical Analysis

Experimental data were analyzed with two-tailed Student’s *t*-test (**, *p* < 0.01; ***, *p* < 0.001) using GraphPad Prism 8.2.1 for intergroup comparison difference. The data were presented as means ± standard errors of the mean (SEM).

## 3. Results

### 3.1. Identification and Characterization of the Intersex Homologues in BPH

To identify the *intersex* homologues in BPH, we used the known *D. melanogaster intersex* protein as a tblastn query sequence for both the BPH genome and transcriptome database. Eventually, we found one *Nlix-*Like locus in the genome and two transcripts in the transcriptome. 

By comparing the corresponding genomic and transcripts sequences with the two transcripts, we found that the two transcripts were from alternative splicing of one gene. The PCR results showed that the two transcripts were expressed in both sexes. Thus, they were named *NlixL*ong (*Nlix*^L^) and *NlixShort*(*Nlix*^S^), respectively (Figure 1C). Comparison of the two transcripts with the corresponding genome sequences revealed that they shared the first exon. The *NlIX*^S^ transcript was generated by introducing an alternative translation stop codon in the first exon, leading to the production of a truncated protein (Figure 1A), which might be a non-functional form without the highly conserved region (Med29). Our structure analysis of the *Nlix*^L^ sequence revealed a 558 bp long open reading frame encoding a 186 amino acid polypeptide. The NlIX^L^ protein had the typical characteristic structure of Intersex with several stretches of amino acid residues that were highly conserved only among the IX homologs at the C-terminal. In addition, a region of about 47 residues was identified at the NlIX^L^ N-terminal, which presented glutamine-, methionine-, proline-, and glycine-rich regions in IXs (Figure 1B). In *D. melanogaster*, the DmIX contained transcriptionally active domain in the N-terminal region and several stretches of amino acids at the C-terminal that were crucial for binding DSX^F^ to regulate somatic sexual differentiation [17].

### 3.2. Phylogenetic Analysis

By searching the NCBI database, we identified putative homologs of IX proteins from four Coleoptera, five Hymenoptera, five Lepidoptera, five Diptera, three Blattaria, two Thysanoptera and seven Hemiptera species for investigating the evolutionary relationships of different insects. Phylogenetic analysis showed that these IX proteins were clustered within each insect order, which was consistent with the evolutionary relationship of taxonomic orders (Figure 2). However, the NlIX protein was closely grouped with BtIX from *B. tabaci,* a result inconsistent with the evolutionary relationships of taxonomic Hemiptera suborders. In addition, IX proteins from the homometabolous hymenopteran insects exhibited a closer relationship with hemimetabolous blattarian insects, another result inconsistent with the evolutionary relationships based on genomes in Insecta, indicating IX evolution does not reflect taxonomic relationships.

### 3.3. Temporal and Spatial Expression Patterns of Nlix

To better understand the potential function of *ix* homologs in BPHs, the temporal and spatial expression patterns of *Nlix* were investigated by real-time quantitative PCR (qPCR). Total RNAs were extracted from various samples, including eggs, nymphs of five instars, as well as female and male adults that emerged in 24 h, in order to investigate the developmental expression patterns of BPHs. Our results revealed that *Nlix* was highly abundant in newly laid eggs, at which point the embryos were just beginning to develop. The *Nlix* expression then decreased sharply during the early 6-12 h of embryo stage and thereafter remained stable at a low level (Figure 3B). These suggest that *Nlix* mRNA in the newly laid eggs may be maternally inherited and play specific roles in the early embryonic development of fertilized eggs.

Regarding the tissue-specific expression pattern analysis of *Nlix*, the qPCR results showed a high transcript level in the ovary, but a low transcript level in a fat body, gut, testis and integument (Figure 3A), indicating that this gene may play an important function in the development of the female reproductive system and/or in embryogenesis.

### 3.4. Phenotypes and RNAi Effects

To verify the possible function of *Nlix*, the 2nd, 3rd and 4th instar nymphs were injected with double-stranded RNA (dsRNA). Moreover, we chose two non-overlapping *Nlix* regions to perform independent RNAi experiments in nymphs, thereby excluding off-target effects. A similar phenotype was observed upon injection of dsRNAs targeting two different regions, suggesting that no off-target effects occurred in the RNAi experiments. As mentioned above, since *Nlix*^S^ had the potential to encode a truncated non-functional protein and had only a 24 bp differential sequence compared to *Nlix*^L^, we did not find any *Nlix*^S^-specific RNAi fragment. Hence, we primarily investigated the function of *Nlix*^L^ in this study. RT-PCR analysis revealed that ds*Nlix* was capable of suppressing the *Nlix* expression efficiently (Figure 4D). 

As shown in Figure 4, injection of dsRNA for *Nlix* in 2nd, 3rd and 4th instar nymphs led to lethal phenotypes at different stages. Remarkable decreases in mortality were observed when these instar nymphs were treated with ds*Nlix*, showing 1.3% (2nd instar), 6.1% (3rd instar) and 24.7% (4th instar) survival rates 8 days after injection, which differed significantly from the ds*GFP* treated nymphs (Figure 4A–C). In addition, all dead BPHs exhibited the same phenotype, which failed to shed their old cuticle during the nymph-nymph molting or the nymph-adult ecdysis stage (Figure 4E). This suggests that *Nlix* is essential for the development of BPH.

### 3.5. Nlix Influences Somatic Development of Females and Males

In BPH, normal sex-specific external characteristics comprise aedeagus and harpagones in male and ovipositor in female (Figure 5A,B). Injection of 3rd-instar BPH nymphs with ds*Nlix* caused high mortality, and only a few individual males successfully emerged, while all females died. In these surviving males, the length of aedeagus and claspers were significantly reduced compared to the ds*GFP* treated males (Figure 5E). Silencing of *ix* expression in 4th-instar nymphs produced defects in both sexes, deforming the morphology of ovipositor in females and reducing the size of aedeagus and harpagones in males (Figure 5C,D). In addition, the BPH males treated with ds*Nlix* presented small and abnormal wings when injected into 3rd- and 4th-instar nymphs. For ds*Nlix*-treated females, no obvious difference in wings was observed (Appendix A). 

### 3.6. Effects of Nlix on BPH Fertility and Embryogenesis

When ds*Nlix* treatment was performed during 3rd and 4th instar nymph stages, the BPH adults were sterile due to abnormal development of genital structures, which were essential for adult courtship and mating. These deformed males and females were dissected four days after emergence, and results showed that the ovaries were poorly developed, while the testes had no obvious morphological differences compared to the ds*GFP*-treated group (Figure 6A,B,D,E,G). 

As the male testes matured immediately after emergence, we further conducted RNAi assays on late 5th instar males, in order to observe the phenotypes of testicular and embryonic developments in the offspring. The testes developed normally and produced healthy sperm (Figure 6D,F), and no significant difference in eggs production or hatchability was observed when ds*GFP*-and ds*Nlix*-treated males mated with wild-type virgin females (Figure 7A,B).

According to the temporal expression profiles, the expression level of the *ix* gene was particularly high during the early stage of embryogenesis. Hence, we performed RNAi experiments on the newly emerged females without full ovarian development, in order to observe the effects of *Nlix* on female fertility and embryogenesis. As shown in Figure 6A,C, the ovaries of the mated females four days after injection were normal with developed seminal receptacles, lateral oviducts and oocytes in both the control and treatment groups. Nevertheless, the dysplastic copulatory bursas of ds*Nlix*-treated females were abnormally enlarged compared to the ds*GFP*-treated females. After the knockdown of *Nlix* gene, the female fecundity was seriously decreased. Results based on oviposition and hatchability experiments demonstrated that the ds*Nlix*-treated females laid only an average of 19 eggs in 5 days, and all of the eggs failed to hatch and died in leaf sheaths. On the contrary, approximately 100 eggs with a 93.1% hatchability were observed in the control group (Figure 7A,B). these results suggest that without the expression of *Nlix*, the eggs cannot develop normally into 1st instar nymph. The eye pigmentation, a characteristic maker of normal embryo development, failed to be formed and generally appeared at about 5 days after oviposition (Figure 7D). These results indicated that the *Nlix* is indispensable for the development of the reproductive system and offspring embryogenesis in BPH females. 

### 3.7. Nlix Functions Independently of Nldsx 

In the fly *D. melanogaster*, the IX and DSX^F^ proteins function together to promote the expression of yolk protein genes and to control the somatic sexual differentiation at the bottom of the sex-determining hierarchy in females [1]. However, depletion of *Nldsx^F^* only led to larger body size with normal abdomen structure and fertility in females, while knockdown of *Nldsx^M^* resulted in feminizing phenotype with abnormal gonads and external genital structure in males [13], which differed strikingly from the phenotype characteristics of silencing *Nlix* in BPHs (Figure 5 and Figure 6). As *Nldsx* and *Nlix* exhibited some roles different from those in *D. melanogaster*, we further evaluated the transcript level of *Nldsx* gene in the *dsNlix*-treated BPHs and the expression level of *Nlix* gene in the *dsNlix*-treated BPHs to see if the two genes affect each other at the transcriptional level. The analysis revealed that the RNAi-mediated *Nlix* knockdown in the whole body did not affect the splicing transcript level of *Nldsx* gene and vice versa (Figure 8). These results suggested that *Nlix* might function independently of *Nldsx* at the transcriptional level. 

## 4. Discussion

Genes terminally positioned in sex-determining cascades, such as the *dsx* and *ix* genes in the sex-determining pathway, are broadly conserved and implement sexually dimorphic differentiation in insect taxa. In this paper, we identified the *ix* gene and characterized its biological functions in BPH. The RNAi experimentation was performed successfully and the result suggested pleiotropism of *Nlix* in BPH.

Sequence analysis revealed that the *Nlix* encodes a conserved domain, namely, the mediator complex subunit 29 (Med29), as reported in other insect species. The mediator, asa large multiprotein complex that contains Med29 subunit in many species, acts as a transcriptional coactivator and is conserved in eukaryotes from yeast to humans, which is required for transcriptional activation of RNA polymerase II that interacts with DNA-binding transcription factors and general initiation activators, thereby regulating the messenger RNA synthesis of target genes [18,19]. In *D. melanogaster*, an expected transcription factor DmIX, acts in concert with DmDSX^F^, a DNA-binding protein, to regulate the expression of the yolk protein genes [1]. One may therefore expect that *Nlix* is well conserved in BPH, which functions with DNA-bound transcriptional activators to participate in many biological processes.

As is known, sexual differentiation is achieved by the sex-specific alternative splicing of a range of genes involved in the sex-determining hierarchy. Nevertheless, the *Dmix* is transcribed non-sex specifically and contains only one exon in *D. melanogaster* [1]. Despite discoveries of sex-specific splicing transcripts of *ix* in many insects, e.g., a testis-specific isoform of *ix* founded in *B. mori* [20], female pupae stage-specific *ix* isoform described in *M. vitrata* [3] and two sex-specific variants of *ix* reported in *B. tabaci* [5], the underlying functions of these sex-specific isoforms remain largely unknown. In BPH, the *Nlix* produces two isoforms by alternative splicing, both of which are expressed non-sex specifically. For *Nlix^S^*, the shorter one of the two transcripts, *we* failed to evaluate its biological function separately in the RNAi experiment due to the excessively short sequence used to distinguish it from the longer. Thus, the function of *Nlix^S^* in BPH needs further investigation. Nevertheless, *Nlix^S^* may be only a non-functional isoform derived from *Nlix* regulation, as it loses the highly conserved region (Med29).

In the present study, we observed high lethality in BPH nymphs after treatment with ds*Nlix* in the early nymph stage, and the survival rate increased when the dsRNA injection was chosen in later instars. A similar case was also described in the *Nltra2* gene of BPH, and the injection of ds*Nltra-2* in earlier instars of female nymphs resulted in more efficient masculinization of females [11]. This effect is probably attributed to the accumulated influence of *Nlix* over the growth period in BPH. 

The *ix* gene plays an essential role in the normal development of female external genital structure in different insects. Accordingly, we focused on the external genital characteristics of ds*Nlix*-treated BPHadults in this study. In *D. melanogaster*, the female terminalia of *Dmix* mutant individuals exhibited intersexual phenotypes with a fewer number of vaginal teeth, fused dorsal lateral anal plates and an increased number of abnormal foreleg bristles. In contrast, among male *Dmix* mutants, no effects in somatic sexual phenotypes were detected [1,8]. Studies in *B. mori* and *C. metallifer* have similarly shown that the *ix* gene was indispensable for normal external genital development of female insects, but not for male individuals [2,4]. Our results demonstrated that after ds*Nlix* microinjection during the nymph stage (4th instar), the external genitalia of both male and female BPH adults displayed a defective phenotype, which led to a reproductive loss in both sexes. This observation suggests that *Nlix* not only contributes to the female somatic sexual development but is also required for male normal terminalia in this hemimetabolous insect, a phenomenon quite different from that in the holometabolous insects.

Although previous studies have provided some insights into the *ix* gene functions on gonads, the relevant detailed description remains scarce. In *D. melanogaster*, severe reduction of internal genitalia was found in female *Dmix* mutants, which were attributed to cell death in the genital disc [8]. Research on *B. mori* showed that the internal reproductive organs of female or male *Bmix* mutant individuals developed normally [2]. In our study, ds*Nlix* treatment produced no visible effects on the male gonad development among late 5th instar nymphs. Nevertheless, the *Nlix* knockdown mediated by RNAi experiment at 4th nymphal instar led to arrested ovary development in BPH females, suggesting that the *Nlix* action in the nymph stage is required for the development of the female reproductive system. Most interesting to us, the copulatory bursas, a component of the female reproductive system, exhibited marked neoplastic enlargement after injection of ds*Nlix* into the newly emerged females, while the oocytes developed normally, showing a banana-like shape. This phenotype of copulatory bursa malformation, which was attributed to the functional loss of *ix* gene was first discovered in insects, which might account for the dramatic decrease in oviposition among ds*Nlix*-treated females. Our results indicate that *Nlix* may play indispensable roles in many biological activities that occurred in copulatory bursas during female spawning. Further research is needed to clarify the mechanism underlying the contribution of *Nlix* to oviposition.

## 5. Conclusions

In summary, we cloned and characterized *ix* homologs in BPH, and found that *Nlix* contains two isoforms by alternative splicing, both of which are ubiquitously expressed non-sex specifically. Sequence and phylogenetic analyses suggest that the *ix* gene sequences are well-conserved in insects. The RNAi-mediated knockdown of *Nlix* reveals its vital role in the development of external genitalia in both male and female individuals and of internal genitalia in female insects. Additionally, the results of this study also indicate that *Nlix* is required during nymphal and early embryonic development. Most notably *Nlix* is found to participate in the biological processes of copulatory bursas. These findings demonstrated the pleiotropic functions of *Nlix* in *N. lugens*, which will facilitate our understanding of the evolutionary relationship of *ix* gene in Insecta and provide a potential target for the effective control of BPH. Future research will focus on discovering the underlying mechanism of copulatory bursa enlargement induced in the female internal genitalia in the absence of *Nlix* function.

## Figures and Tables

**Figure 1 genes-12-00379-f001:**
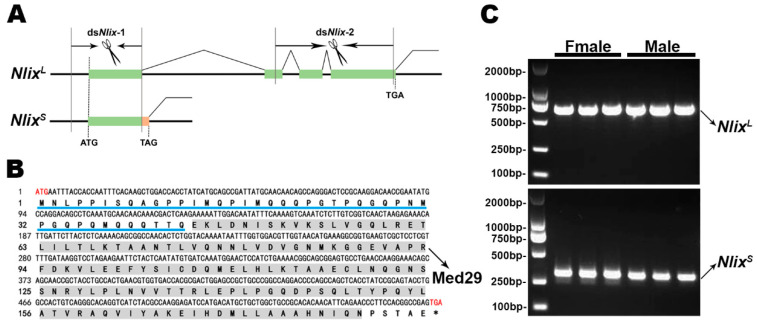
Alternative splicing pattern of *Nlix*. (**A**) Structural schematic of the two *Nlix* isoforms. Boxes and lines denote exons and introns, respectively. Green boxes: common region of the two isoforms and the specific region of *Nlix^L^*. Orange box: the specific region of *Nlix^S^*. Scissors indicate two regions as RNAi targets on the *Nlix^L^*. (**B**) cDNA and corresponding amino acid sequence of *Nlix*. The glutamine-, methionine-, proline-, and glycine-rich regions are underlined and the highly conserved region (Med29) of IX homologs is highlighted in gray. (**C**) Non-sex specific expression of the two *Nlix* splicing variants.

**Figure 2 genes-12-00379-f002:**
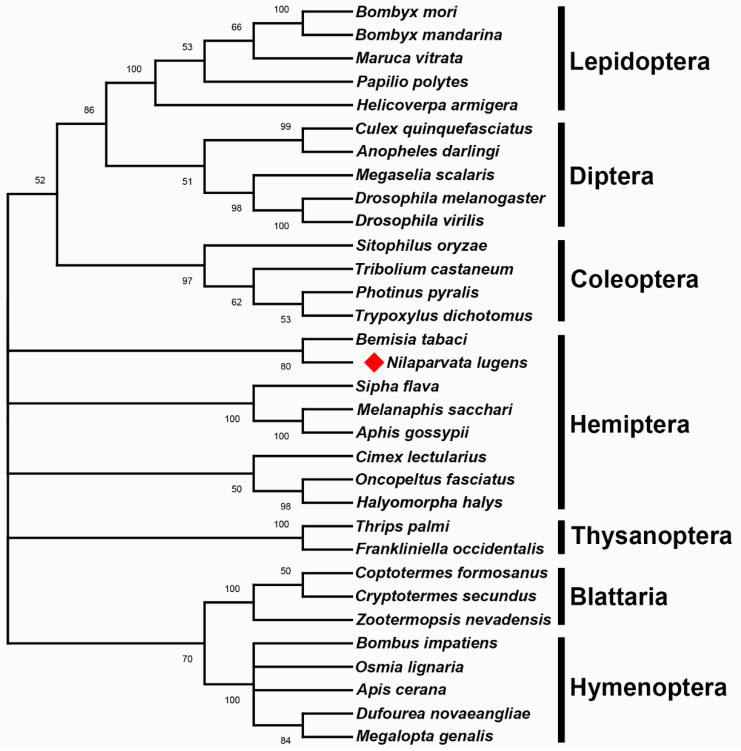
Phylogenetic analysis of the IX homologs. The phylogenetic tree was constructed based on IX proteins from 32 species using the neighbor-joining algorithm. Bootstrap values are shown at the nodes, whereas the NlIX is shown with a red rhombus.

**Figure 3 genes-12-00379-f003:**
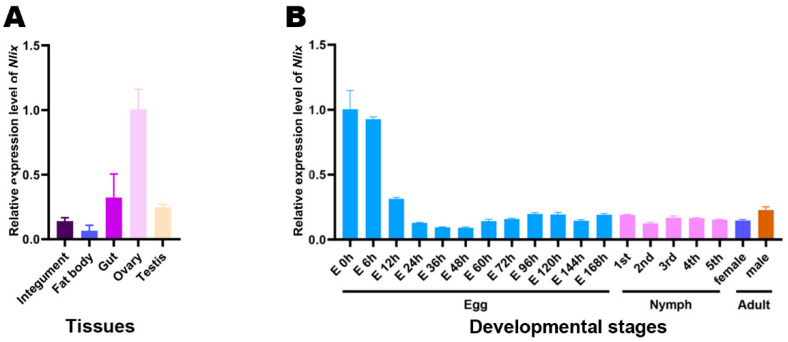
Temporal and spatial expression patterns of *Nlix*. (**A**) Tissue expression patterns of *Nlix*. Tissue samples were dissected from 50 adults 72 h after emergence, including integument, fat body, gut, ovary and testis. (**B**) Developmental stage expression pattern of *Nlix*. Total RNAs were extracted from whole insects at all life stages of BPH. Samples were collected every 24 h from the beginning of each stage. *Nl18S* was used to normalize the transcript level of *Nlix* for qPCR.

**Figure 4 genes-12-00379-f004:**
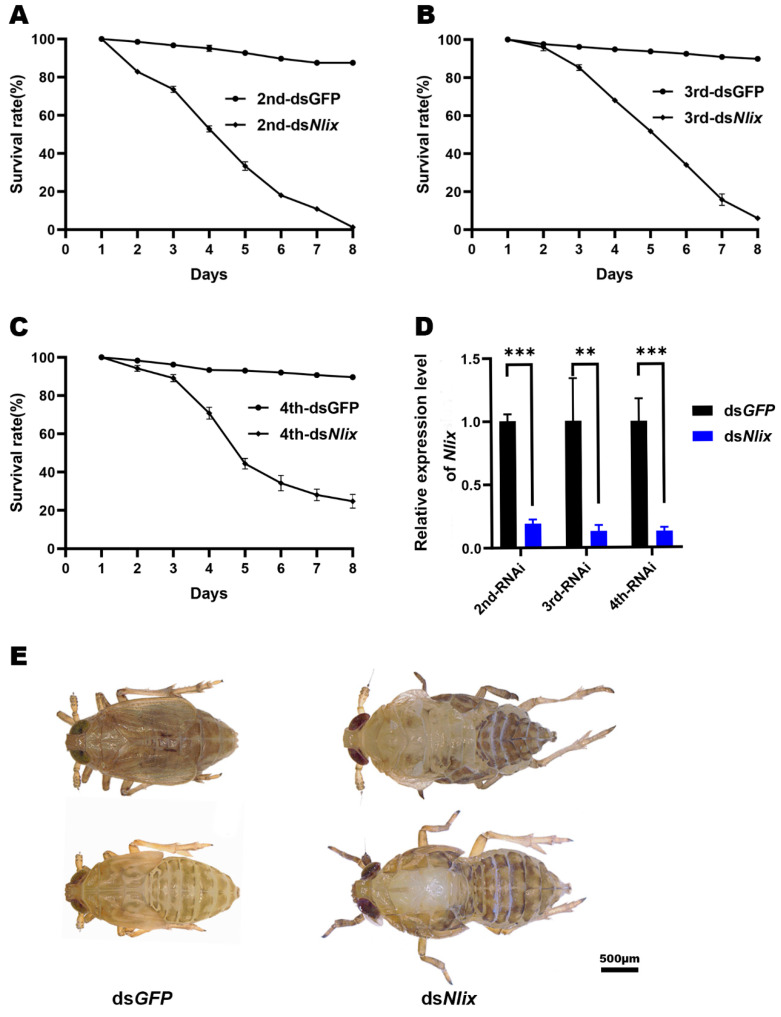
Effects of *dsNlix* treatment on nymph development. (**A**–**C**) Survival rates after the knockdown of *Nlix*. ds*GFP* was used as the negative control in each group. (**A**) 2nd instar nymphs after dsRNA treatment; (**B**) 3rd instar nymphs after dsRNA treatment;(**C**) 4th instar nymphs after dsRNA treatment. (**D**) Transcript levels of Nlix after RNAi treatment in the 2nd, 3rd and 4th instar nymphs. The RNAi efficiency of *Nlix* was evaluated by collecting 6 to 10 insects three days after injection. Means ± SEM from three experiments. ** *p* < 0.01, *** *p* < 0.001 (Student’s *t*-test). (**E**) The lethal phenotype of *N. lugens* injected with ds*Nlix* in nymph stages. Most nymphs die of molting failure.

**Figure 5 genes-12-00379-f005:**
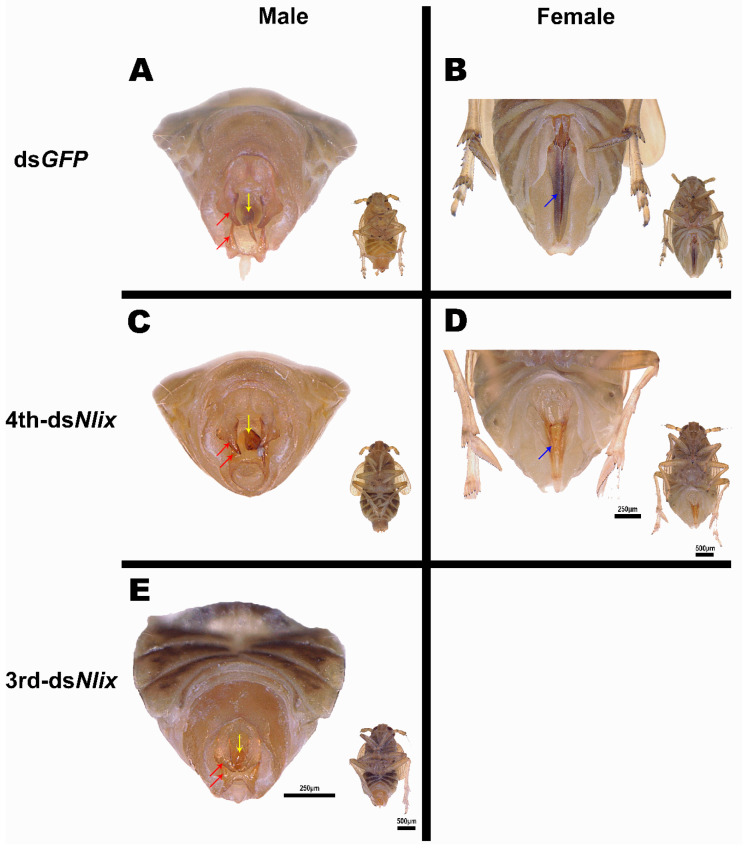
The external genitalia of BPH adults injected with ds*GFP* and ds*Nlix*. (**A**,**B**) The abdomen of female BPHs treated with ds*GFP* and ds*Nlix* for 4th instar nymphs. The ovipositors of ds*Nlix*-treated females were deformed. (**C**–**E**) The abdomen of male BPHs treated with ds*GFP* and ds*Nlix* for 3rd and 4th instar nymphs. The size of external genitalia was reduced in ds*Nlix*-treated males. The red arrow: harpagones; the yellow arrow: aedeagus; and the blue arrow: ovipositor. Images at the bottom right are the full images of treated BPH.

**Figure 6 genes-12-00379-f006:**
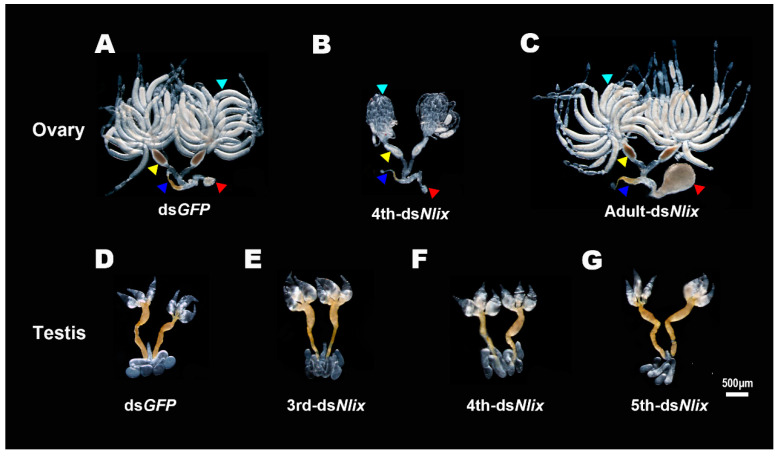
Phenotypes of gonad development upon *Nlix* knockdown in females or males. (**A**–**C**) The ovaries of BPH females treated with ds*GFP*, and ds*Nlix* for 4th instar nymphs and newly emerged adults. Red triangles: copulatory bursas; blue triangles: seminal receptacles, yellow triangles: lateral oviducts; and cyan triangles: oocytes. (**D**–**G**) The testes of BPH males were treated with ds*GFP* and ds*Nlix* for 3rd, 4th and 5th instar nymphs, respectively. ds*GFP* was injected as a negative control for the nonspecific effects of dsRNA. These internal reproductive systems were dissected from adults 4 days after eclosion.

**Figure 7 genes-12-00379-f007:**
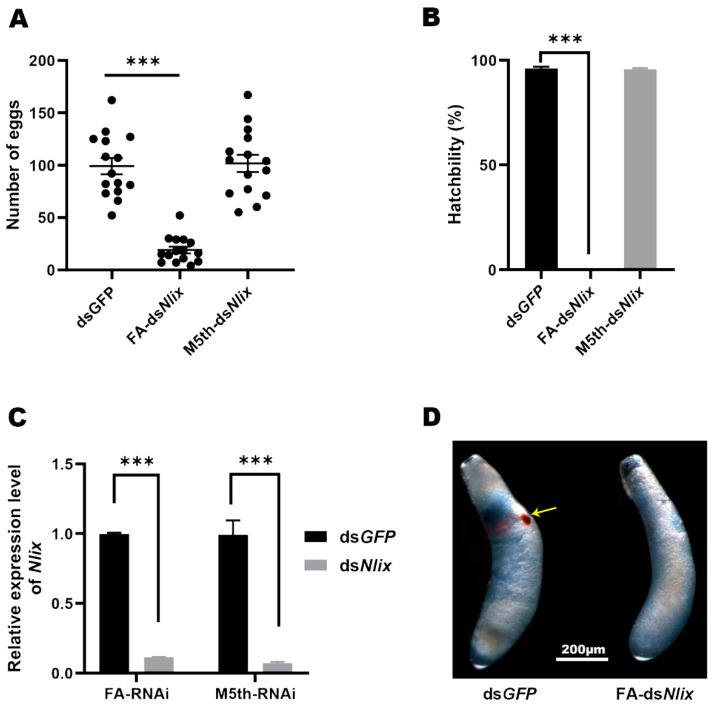
*Nlix* was indispensable for female fecundity and embryonic development. (**A**) The offspring number of ds*Nlix*-treated females (FA-ds*Nlix*) after mating with ds*GFP*-treated males, or the number of eggs laid by the ds*GFP*-treated females after mating with the males treated with ds*Nlix* at 5th instar (M5th-ds*Nlix*). ds*GFP* was injected as a negative control. The newly emerged female or male individuals were injected with dsRNA. (**B**) The egg hatching rate after RNAi treatment. Fifteen replications were administered. (**C**) The transcript level of *Nlix* significantly decreased after dsRNA treatment. The results are shown as the mean ± SEM. Student’s *t*-test was used to compare the differences from ds*GFP* (*** *p* < 0.001). (**D**) The embryonic development after knockdown of *Nlix*. The red eye pigmentation was indicated by the yellow arrow.

**Figure 8 genes-12-00379-f008:**
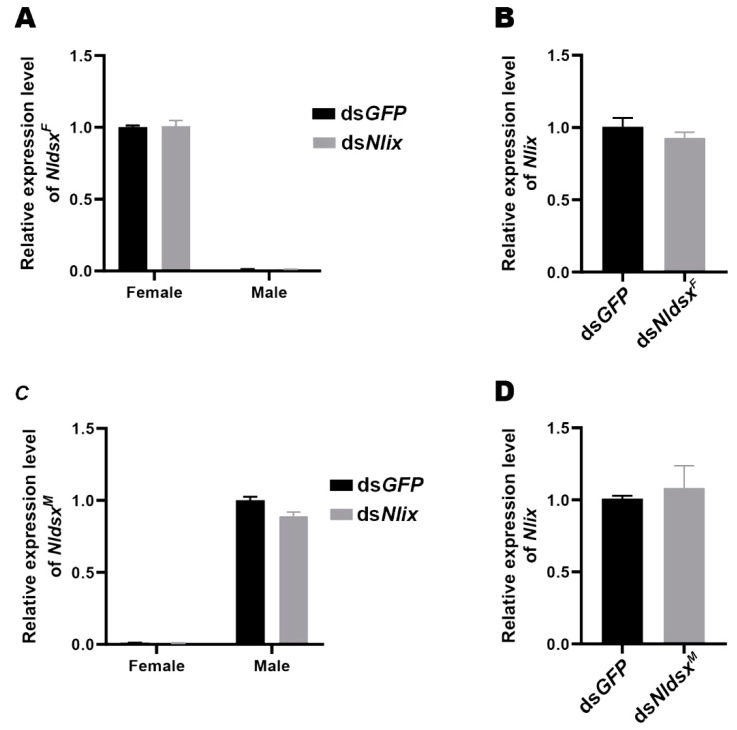
No correlation was found regarding the expression level between *Nlix* and *Nldsx*. (**A**,**C**) Knockdown of *Nlix* produced no obvious effect on the level of female-specific isoform *Nldsx^F^* or male-specific isoform *Nldsx^M^*. (**B**,**D**) The *Nlix* expression was unaffected in the ds*Nldsx^F^*-treated females or ds*Nldsx^M^*-treated males. The values were calculated from three biological replications (mean ± SEM).

## Data Availability

The brown planthopper *intersex* gene sequence is doposited in the GenBank with the accession number: XP_022205962.1.

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
