# Peer review of "Pleiotropic Roles of the Orthologue of the *Drosophila melanogaster Intersex* Gene in the Brown Planthopper"

_genes, 2021, doi:10.3390/genes12030379_

Round 1
Reviewer 1 Report
This study presents well done research on an ortholog of the intersex gene, which is known to be involved in the sex-determining cascade in Drosophila melanogaster, identified in the a serious pest of rice, the brown planthopper (Nilaparvata lugens). The authors showed that the N. lugens intersex gene (Nlix) is expressed in both sexes and does not play a role in sex determination. Using RNAi silencing of the gene, they convincingly showed that the gene is vital for the embryonic and nymph development, for the development of external genitalia in both females and males, and for the development of internal reproductive organs in females. Methods use are very well described, results obtained are well supported by figures, and conclusions made are convincing. Thus, the study represents a significant contribution for the understanding of sexual development in hemimetabolous insects and deserves publishing. However, two issues prevent me from recommending the acceptance of this article for publication. First is specified below in my “Specific comment. The second is English, which requires deep revision, either by a English native speaker or by an English editing service. I also made a number of suggestions for minor revisions of the text (see the list below).
Specific comment
Sections 2.4. and 3.3. Why the 18S ribosomal RNA gene (Nl18S) from the multigene rDNA family was used to normalize Nlix expression patterns? This gene occurs in many copies at unknown numbers (>100 copies) in the genome. It is not a housekeeping gene, as the authors claim, and this multi-copy gene is not suitable as a reference gene for a single-copy gene such as Nlix. The authors should either remake the RT-qPCR expression experiments with a suitable single-copy housekeeping gene as a reference or they should convincingly explain why they used the 18S ribosomal RNA gene, and add a supplementary figure showing a comparison of the expression patterns (e.g. RT-qPCR curves) of the 18S RNA gene and the Nlix gene.
Minor suggestions
Line 2/3: Multifunctional Roles of the Sex-determining Cascade Gene Intersex in the Brown Planthopper
Line 16: omit “the” before Drosophila
Line 19: ix should be written in italics
Line 20: omit “and” in “variants and expressed in both”
Line 42: Nix should be written in italics and with capital “N”
Line 43: Yob should be written in italics
Line 46: full name is (Fem piRNA) and Fem should be italicized
Line 46: correct “doulesex” to “doublesex”
Line 52: The intersex (ix) gene …
Line 68/69: … two non-sex-specific variants that contrast with ix of D. melanogaster, which is not spliced …
Line 69: … experiments in the whitefly …
Line 74: In BPH, details of the molecular mechanism …
Line 79: … including the development of gonad, …
Line 82: … role in the development …
Line 84: … most sperm in the testes was inactive.
Line 86: Sex-lethal protein [as protein, it should be written in regular fonts with capital „S“]
Line 87: TRA2 protein
Line 89: Sxl homolog
Line 93/94: ... loss of dsx function resulted in feminization of BPH males [11, 22].
Line 95: ... to further explore ...
Line 101: ... population used in this paper was originally collected ...
Line 104: Searching for ix homologs in BPH ...
Line 129: ... which included ...
Line 130: 24 h
Line 132: 72 h
Line 133: 1 μg
Line 134: 10 μL
Line 135: ... manufacturer’s instructions.
Line 145: Aequorea victoria
Line 146: ... are shown ...
Line 149: 10 nL of dsRNA (5 ng/μL)
Line 152: three days after injection
Line 157: three days after emergence
Line 160: Afterward
Line 164: Student’s t-test [“t” should be written in italics]; “P” as probability should be written in italics
Line 180: 558 bp
Line 181: 20.5 kDa
Line 199: ... by searching in the NCBI database ...
Lines 201 and 205: relationships
Line 206: ..., indicating that IX evolution does not reflect taxonomic relationships.
Line 208/209: based on IX proteins
Line 212: if gene, than “intersex” should be written in italics; if protein, then “Intersex”
Line 215: 24 h
Line 216: ... that Nlix was highly expressed ...
Line 217: 6-12 h
Line 218: ... then remained stable ...
Line 220/221: ... showed a high transcript level in the ovary, but a low transcript level in the fat body, ...
Line 226: 72 h
Line 228: ... of BPH.
Line 236: 24 bp
Line 251: 6 to 10 insects
Lines 253 and 314/315: “P” as probability should be written in italics and by capital letter; “t” in Student’s t-test should be written in italics
Line 318 and 346: the yolk protein
Line 326: … did not affect …
Line 327: (Figure 8)
Line 349: As is known, sexual differentiation is achieved …
Line 355: … isoforms remain largely unknown.
Line 357: … conducted in the RNAi experiment …
Line 363/364: It is very difficult to understand the following phrase: “… the females became more efficient masculinization resulted from the earlier instars injected with dsNlTra-2[21].” Please re-phrase it.
Line 369: the female terminalia
Line 371: no effects
Line 380: … on gonads, …
Line 400: external genitalia
Line 401: internal genitalia
Line 406: in Insecta
Line 407: will focus
Line 408: … internal genitalia in the absence …
Line 425: csd should be written in italics
Line 430: Mdmd and CWC22 should be written in italics
Line 430/431: omit “(American Association for the 430 Advancement of Science)”
Line 438: Maleness-on-the-Y (MoY) – both the gene name and gene abbreviation should be written in italics
Line 439/440: omit “(American Association for the 430 Advancement of Science)”
Lines 444 and 469: Doublesex should be written in italics
Line 446/447: Phylogenetic distribution and evolutionary dynamics of the sex determination genes doublesex and transformer in insects. [use small initial letters in the title words; doublesex and transformer should be written in italics]
Lines 448, 452, 460, 463, 466, 472, 480, 484, 488: doublesex should be written in italics
Line 455: Ceratitis capitata should be written in italics
Lines 457 and 483: transformer should be written in italics
Line 466: Sex-specific splicing
Line 467: insect sex-determination pathway
Line 476: Tra-2 should be written in italics
Line 477: cross-talk with small initial letters; Nilaparvata with capital “N”
Line 482: Sex-lethal should be written in italics and with capital “S”
Line 485: transformer and transformer-2 should be written in italics
Line 486: intersex should be written in italics and small initial letter
Line 489: Intersex should be written in italics
Lines 492, 498, 502, 507: intersex should be written in italics
Line 498: Bemisia should be written with capital “B”
Line 504: Intersex (ix) –both the gene name and gene abbreviation should be written in italics
Line 506: dpp should be written in italics
Line 514/515: Proc Nat Acad Sci U S A
Line 516/517: Doublesex and the regulation of sexual dimorphism in Drosophila melanogaster.
Line 523: intersex should be written in italics
Line 537: … dsRNA.
Author Response
Reviewer 1. This study presents well done research on an ortholog of the intersex gene, which is known to be involved in the sex-determining cascade in Drosophila melanogaster, identified in the a serious pest of rice, the brown planthopper (Nilaparvata lugens). The authors showed that the N. lugens intersex gene (Nlix) is expressed in both sexes and does not play a role in sex determination. Using RNAi silencing of the gene, they convincingly showed that the gene is vital for the embryonic and nymph development, for the development of external genitalia in both females and males, and for the development of internal reproductive organs in females. Methods use are very well described, results obtained are well supported by figures, and conclusions made are convincing. Thus, the study represents a significant contribution for the understanding of sexual development in hemimetabolous insects and deserves publishing. However, two issues prevent me from recommending the acceptance of this article for publication. First is specified below in my “Specific comment. The second is English, which requires deep revision, either by a English native speaker or by an English editing service. I also made a number of suggestions for minor revisions of the text (see the list below). Author response: We greatly appreciate the reviewer for his/her positive comments and suggestions. We have had an English editing service to polish the English usage. All other minor revisions were revised as the reviewer’s suggestions. Specific comment Sections 2.4. and 3.3. Why the 18S ribosomal RNA gene (Nl18S) from the multigene rDNA family was used to normalize Nlix expression patterns? This gene occurs in many copies at unknown numbers (>100 copies) in the genome. It is not a housekeeping gene, as the authors claim, and this multi-copy gene is not suitable as a reference gene for a single-copy gene such as Nlix. The authors should either remake the RT-qPCR expression experiments with a suitable single-copy housekeeping gene as a reference or they should convincingly explain why they used the 18S ribosomal RNA gene, and add a supplementary figure showing a comparison of the expression patterns (e.g. RT-qPCR curves) of the 18S RNA gene and the Nlix gene. Author response: Thank you for the suggestion. 18S ribosomal RNA gene was extensively used as internal reference gene in planthoppers and many insects. 18S ribosomal RNA has been considered as an ideal reference gene due to its apparent relatively invariable rRNA expression levels with respect to other genes [Bustin, SA,2000].18S rRNA was found to be one of the most suitable housekeepers in the different developmental stages of Lucilia cuprina [Bagnall NH, Kotze AC, 2010], in different organs of Rhodnius prolixus under diverse conditions [Majerowicz D et al, 2011, Paim RM et al, 2012], and in the planthopper Delphacodes kuscheli [Maroniche et al ,2011]. Though Yuan et al (2014) demonstrated that RPS15was the most suitable referent gene under different temperature and diet conditions, while RPS11 was the most suitable gene under different pesticide exposure and starvation conditions in the planthopper. However, their results also showed 18S ribosomal RNA and ACT were expressed at highest level, and the programs, Delta Ct method, RefFinder and geNorm,identified 18S, RPS11, and RPS15 as the most stable genes in different planthopper body parts. According to geNorm, four reference genes (18S, RPS15, TUB, and EF) should be required for a suitable normalization in the different developmental stages (Yuan et al (2014). In the present paper, we use the reference gene to normalize the ix gene expression level only in different tissues and development stages under normal environmental condition, so we chose the highly expressed18S as the internal reference gene. Anyway, we thank the reviewers this kind reminding, when we perform qPCR for this insect under different stress in the future other research, we will carefully select more other suitable genes as internal reference. References 1. Bustin SA (2000) Absolute quantification of mRNA using real-time reversetranscription polymerase chain reaction assays. J Mol Endocrinol 25: 169–193. 2. Bagnall NH, Kotze AC (2010) Evaluation of reference genes for real-time PCR quantification of gene expression in the Australian sheep blowfly, Lucilia cuprina. Med Vet Entomol 24: 176–181. 3. Majerowicz D, Alves-Bezerra M, Logullo R, Fonseca-de-Souza AL, Meyer-Fernandes JR, et al. (2011) Looking for reference genes for real-time quantitative PCR experiments in Rhodnius prolixus (Hemiptera: Reduviidae). Insect Mol Biol 20(6): 713–722. 4. Paim RM, Pereira MH, Ponzio RD, Rodrigues JO, Guarneri AA, et al. (2012) Validation of reference genes for expression analysis in the salivary gland and the intestine of Rhodnius prolixus (Hemiptera, Reduviidae) under different experimental conditions by quantitative real-time PCR. BMC Research Notes 5: 128. 5. Maroniche GA, Sagadı´n M, Mongelli VC, Truol GAM, del Vas M (2011)Reference gene selection for gene expression studies using RT-qPCR in virusinfected planthoppers. Virology J 8: 308–315. 6. Yuan M, Lu Y, Zhu X, Wan H, Shakeel M, et al. (2014) Selection and Evaluation of Potential Reference Genes for Gene Expression Analysis in the Brown Planthopper, Nilaparvata lugens (Hemiptera: Delphacidae) Using Reverse-Transcription Quantitative PCR. PLoS ONE 9(1): e86503. doi:10.1371/journal.pone.0086503 Minor suggestions Line 2/3: Multifunctional Roles of the Sex-determining Cascade Gene Intersex in the Brown Planthopper Line 16: omit “the” before Drosophila Line 19: ix should be written in italics Line 20: omit “and” in “variants and expressed in both” Line 42: Nix should be written in italics and with capital “N” Line 43: Yob should be written in italics Line 46: full name is (Fem piRNA) and Fem should be italicized Line 46: correct “doulesex” to “doublesex” Line 52: The intersex (ix) gene … Line 68/69: … two non-sex-specific variants that contrast with ix of D. melanogaster, which is not spliced … Line 69: … experiments in the whitefly … Line 74: In BPH, details of the molecular mechanism … Line 79: … including the development of gonad, … Line 82: … role in the development … Line 84: … most sperm in the testes was inactive. Line 86: Sex-lethal protein [as protein, it should be written in regular fonts with capital „S“] Line 87: TRA2 protein Line 89: Sxl homolog Line 93/94: ... loss of dsx function resulted in feminization of BPH males [11, 22]. Line 95: ... to further explore ... Line 101: ... population used in this paper was originally collected ... Line 104: Searching for ix homologs in BPH ... Line 129: ... which included ... Line 130: 24 h Line 132: 72 h Line 133: 1 μg Line 134: 10 μL Line 135: ... manufacturer’s instructions. Line 145: Aequorea victoria Line 146: ... are shown ... Line 149: 10 nL of dsRNA (5 ng/μL) Line 152: three days after injection Line 157: three days after emergence Line 160: Afterward Line 164: Student’s t-test [“t” should be written in italics]; “P” as probability should be written in italics Line 180: 558 bp Line 181: 20.5 kDa Line 199: ... by searching in the NCBI database ... Lines 201 and 205: relationships Line 206: ..., indicating that IX evolution does not reflect taxonomic relationships. Line 208/209: based on IX proteins Line 212: if gene, than “intersex” should be written in italics; if protein, then “Intersex” Line 215: 24 h Line 216: ... that Nlix was highly expressed ... Line 217: 6-12 h Line 218: ... then remained stable ... Line 220/221: ... showed a high transcript level in the ovary, but a low transcript level in the fat body, ... Line 226: 72 h Line 228: ... of BPH. Line 236: 24 bp Line 251: 6 to 10 insects Lines 253 and 314/315: “P” as probability should be written in italics and by capital letter; “t” in Student’s t-test should be written in italics Line 318 and 346: the yolk protein Line 326: … did not affect … Line 327: (Figure 8) Line 349: As is known, sexual differentiation is achieved … Line 355: … isoforms remain largely unknown. Line 357: … conducted in the RNAi experiment … Line 363/364: It is very difficult to understand the following phrase: “… the females became more efficient masculinization resulted from the earlier instars injected with dsNlTra-2[21].” Please re-phrase it. Line 369: the female terminalia Line 371: no effects Line 380: … on gonads, … Line 400: external genitalia Line 401: internal genitalia Line 406: in Insecta Line 407: will focus Line 408: … internal genitalia in the absence … Line 425: csd should be written in italics Line 430: Mdmd and CWC22 should be written in italics Line 430/431: omit “(American Association for the 430 Advancement of Science)” Line 438: Maleness-on-the-Y (MoY) – both the gene name and gene abbreviation should be written in italics Line 439/440: omit “(American Association for the 430 Advancement of Science)” Lines 444 and 469: Doublesex should be written in italics Line 446/447: Phylogenetic distribution and evolutionary dynamics of the sex determination genes doublesex and transformer in insects. [use small initial letters in the title words; doublesexand transformer should be written in italics] Lines 448, 452, 460, 463, 466, 472, 480, 484, 488: doublesex should be written in italics Line 455: Ceratitis capitata should be written in italics Lines 457 and 483: transformer should be written in italics Line 466: Sex-specific splicing Line 467: insect sex-determination pathway Line 476: Tra-2 should be written in italics Line 477: cross-talk with small initial letters; Nilaparvata with capital “N” Line 482: Sex-lethal should be written in italics and with capital “S” Line 485: transformer and transformer-2 should be written in italics Line 486: intersex should be written in italics and small initial letter Line 489: Intersex should be written in italics Lines 492, 498, 502, 507: intersex should be written in italics Line 498: Bemisia should be written with capital “B” Line 504: Intersex (ix) –both the gene name and gene abbreviation should be written in italics Line 506: dpp should be written in italics Line 514/515: Proc Nat Acad Sci U S A Line 516/517: Doublesex and the regulation of sexual dimorphism in Drosophila melanogaster. Line 523: intersex should be written in italics Line 537: … dsRNA. Author response: We really appreciate the reviewer for the very careful corrections and suggestions. We revised the manuscript following all the suggestions.Reviewer 2 Report
The authors identified and characterised the ix gene of this species on the basis of homology with ix from Drosophila and other insect species. The transcript is alternatively spliced but neither variant is sex-specific. The function of the longer variant was studied with an RNAi approach, injection of designed dsRNA into the mesothorax of nypmphs and young adults. - I am surprised that the injected dsRNA is spread among tissues and internalised in many or most cells of the injected individual, as it apparently does according to the reduced transcript levels shown in Figure 4B.
In contrast to Drosophila, the planthopper ix shows no indication of cooperation with dsx. The authors found various effects of ix knockdown a high rate of mortality, malformation of female and male exxternal genitalia, exuberant growth of the bursa copulatrix, a reduced production of eggs,
and the few eggs laid did not hatch.
Although the authors do not say this expressedly: it looks as if ix does not belong to the proper sex determining mechanism in the leafhopper. It is involved in many developmental activities and, hence, indeed "pleeiotropic" in the classical genetic sense.
The ms has a serious prblem though. The English requires guessing at several and careful correction correction at many places. It is especially irritating in the Discussion.
Some minor points:
The introduction is overlong and not to the point. The reader looses interest after a while.
line 219: maternally 'inherited'?
Figure 4A: where is '4th-dsGFP'?
Figure 5CDE: Even the arrows do not help much to see what the reader is expected to see
Author Response
Reviewer 2. The authors identified and characterised the ix gene of this species on the basis of homology with ix from Drosophila and other insect species. The transcript is alternatively spliced but neither variant is sex-specific. The function of the longer variant was studied with an RNAi approach, injection of designed dsRNA into the mesothorax of nypmphs and young adults. - I am surprised that the injected dsRNA is spread among tissues and internalised in many or most cells of the injected individual, as it apparently does according to the reduced transcript levels shown in Figure 4B. In contrast to Drosophila, the planthopper ix shows no indication of cooperation with dsx. The authors found various effects of ix knockdown a high rate of mortality, malformation of female and male exxternal genitalia, exuberant growth of the bursa copulatrix, a reduced production of eggs, and the few eggs laid did not hatch. Although the authors do not say this expressedly: it looks as if ix does not belong to the proper sex determining mechanism in the leafhopper. It is involved in many developmental activities and, hence, indeed "pleeiotropic" in the classical genetic sense. The ms has a serious prblem though. The English requires guessing at several and careful correction correction at many places. It is especially irritating in the Discussion. Author response: We appreciate the reviewer for his/her positive comments and suggestions. We have had an English editing service to polish the English usage. All other minor revisions were revised as the reviewers’ suggestions. The title was also changed into “Pleiotropic Roles of the Sex-determining Cascade Gene Intersex in the Brown Planthopper” as two reviewer’s suggestions. Some minor points: The introduction is overlong and not to the point. The reader looses interest after a while. Author response: We removed sentences not closely related to intersex gene in the introduction, and focus only on intersex gene line 219: maternally 'inherited'? Author response: We rephrased this word. Figure 4A: where is '4th-dsGFP'? Author response: We made this figure clearer in the revision. Figure 5CDE: Even the arrows do not help much to see what the reader is expected to see Author response: We enlarged these figures for reader to see clearly.Round 2
Reviewer 1 Report
The revised manuscript is significantly improved, including English grammar and style. I am also satisfied with the authors’ explanation of why they used the 18S ribosomal RNA as a reference gene in their RT-qPCR expression experiments. However, I still found a number of minor inaccuracies in the text that require revision (see my suggestions below).
Minor suggestions:
Lines 132 and 139: this long sentence starts with “In this study” and also end with “in this study”. The latter should be removed.
Line 145: correct “female” to “females”
Line 190: correct “Eventully” to “Eventually”
Line 190: correct “Nlix-Like loci” to “Nlix-like locus”
Line 194: correct “(Figure. 1C)” to “(Figure 1C)”
Line 197: (Figure 1A)
Line 199: correct “Open Reading Frame” to “open reading frame”
Line 204: (Figure 1B)
Line 241: (Figure 3B)
Line 246: (Figure 3A)
Line 253: correct “to normalized” to “to normalize”
Line 261: 24 bp [make space between the value and unit]
Line 264: (Figure 4D)
Line 270: (Figure 4A, B, C)
Line 272: (Figure 4E)
Line 282: (Student’s t-test) [“t” should be written in italics]
Line 287: (Figure 5A, B)
Line 290: (Figure 5E)
Line 292: (Figure 5C, D)
Line 294: (Figure S1)
Line 301: ... harpagones; the yellow arrow: aedeagus; and the blue arrow: ...
Line 309: (Figure 6A, B, D, E, G)
Line 313: (Figure 6D, F)
Line 314: (Figure 7A, B)
Line 320: Figure 6A and C
Line 329: (Figure 7A, B)
Line 329: change “these suggest” to “These results suggest”
Line 333: (Figure 7D)
Lines 338, 339: Red triangles: copulatory bursas; blue triangles: seminal receptacles; yellow triangles: lateral oviducts; and cyan triangles: oocytes.
Line 386: correct “interacst” to interacts”
Line 432: A research in B. mori showed ...
Line 447: correct “clarity” to “clarify”
Line 592: BMC Genomics
Line 611: doublesex should be written in italics
Line 641: Figure S1.
Author Response
Point to point responses to reviewers
Reviewer 1 (Round 2).
The revised manuscript is significantly improved, including English grammar and style. I am also satisfied with the authors’ explanation of why they used the 18S ribosomal RNA as a reference gene in their RT-qPCR expression experiments. However, I still found a number of minor inaccuracies in the text that require revision (see my suggestions below).
Minor suggestions:
Lines 132 and 139: this long sentence starts with “In this study” and also end with “in this study”. The latter should be removed.
Line 145: correct “female” to “females”
Line 190: correct “Eventully” to “Eventually”
Line 190: correct “Nlix-Like loci” to “Nlix-like locus”
Line 194: correct “(Figure. 1C)” to “(Figure 1C)”
Line 197: (Figure 1A)
Line 199: correct “Open Reading Frame” to “open reading frame”
Line 204: (Figure 1B)
Line 241: (Figure 3B)
Line 246: (Figure 3A)
Line 253: correct “to normalized” to “to normalize”
Line 261: 24 bp [make space between the value and unit]
Line 264: (Figure 4D)
Line 270: (Figure 4A, B, C)
Line 272: (Figure 4E)
Line 282: (Student’s t-test) [“t” should be written in italics]
Line 287: (Figure 5A, B)
Line 290: (Figure 5E)
Line 292: (Figure 5C, D)
Line 294: (Figure S1)
Line 301: ... harpagones; the yellow arrow: aedeagus; and the blue arrow: ...
Line 309: (Figure 6A, B, D, E, G)
Line 313: (Figure 6D, F)
Line 314: (Figure 7A, B)
Line 320: Figure 6A and C
Line 329: (Figure 7A, B)
Line 329: change “these suggest” to “These results suggest”
Line 333: (Figure 7D)
Lines 338, 339: Red triangles: copulatory bursas; blue triangles: seminal receptacles; yellow triangles: lateral oviducts; and cyan triangles: oocytes.
Line 386: correct “interacst” to interacts”
Line 432: A research in B. mori showed ...
Line 447: correct “clarity” to “clarify”
Line 592: BMC Genomics
Line 611: doublesex should be written in italics
Line 641: Figure S1.
Author response:
We really appreciate the reviewer for the very careful corrections and suggestions. We revised the manuscript following all the suggestions.
